# Characteristic Human Individual Puffing Profiles Can Generate More TNCO than ISO and Health Canada Regimes on Smoking Machine When the Same Brand Is Smoked

**DOI:** 10.3390/ijerph17093225

**Published:** 2020-05-06

**Authors:** Charlotte G.G.M. Pauwels, Agnes W. Boots, Wouter F. Visser, Jeroen L.A. Pennings, Reinskje Talhout, Frederik-Jan Van Schooten, Antoon Opperhuizen

**Affiliations:** 1Department of Pharmacology and Toxicology, NUTRIM School of Nutrition and Translational Research in Metabolism, Maastricht University, 3581 CD Maastricht, The Netherlands; c.pauwels@maastrichtuniversity.nl (C.G.G.M.P.); A.boots@maastrichtuniversity.nl (A.W.B.); F.vanschooten@maastrichtuniversity.nl (F.-J.V.S.); 2Centre for Health Protection, National Institute for Public Health and the Environment (RIVM), 3721 MA Bilthoven, The Netherlands; Wouter.visser@rivm.nl (W.F.V.); jeroen.pennings@rivm.nl (J.L.A.P.); reinskje.talhout@rivm.nl (R.T.); 3Office of Risk Assessment and Research, Netherlands Food and Consumer Product Safety Authority (NVWA), 3511 GG Utrecht, The Netherlands

**Keywords:** smoking, puffing topography, exposure, TNCO, human study

## Abstract

Human smoking behavior influences exposure to smoke toxicants and is important for risk assessment. In a prospective observational study, the smoking behavior of Marlboro smokers was measured for 36 h. Puff volume, duration, frequency, flow and inter-puff interval were recorded with the portable CReSSmicro™ device, as has often been done by other scientists. However, the use of the CReSSmicro™ device may lead to some registration pitfalls since the method of insertion of the cigarette may influence the data collection. Participants demonstrated consistent individual characteristic puffing behavior over the course of the day, enabling the creation of a personalized puffing profile. These puffing profiles were subsequently used as settings for smoking machine experiments and tar, nicotine and carbon monoxide (TNCO) emissions were generated. The application of human puffing profiles led to TNCO exposures more in the range of Health Canada Intense (HCI)-TNCO emissions than for those of the International Standardization Organization (ISO). Compared to the ISO regime, which applies a low puff volume relative to human smokers, the generation of TNCO may be at least two times higher than when human puffing profiles were applied on the smoking machine. Human smokers showed a higher puffing intensity than HCI and ISO because of higher puffing frequency, which resulted in more puffs per cigarette, than both HCI and ISO.

## 1. Introduction

Around 6.5 million deaths are annually attributable to tobacco smoking worldwide and occur after suffering from tobacco related-diseases [1]. The relationship between tobacco smoking and tobacco related-diseases depends on the quantity and composition of the inhaled cigarette smoke. The adverse health impact of exposure to smoke toxicants is determined by the number of smoked cigarettes, smoking years, cigarette brands, cigarette emissions and smoking topography [2,3,4,5]. How these determinants influence the relationship between smoking and health effects and if they can be used to assess individual health risks require further elucidation [6]. For example, carcinogen intake from mainstream smoke was shown to vary by four-fold between smokers due to inter-individual differences in smoking behavior [7]. In general, however, literature about personal smoking behavior and actual exposure of the lungs to smoke toxicants is scarce [6]. The importance of these individual differences is underlined by the finding of Song et al. who demonstrated a relationship between lung adenocarcinoma and higher smoke volumes in combination with deeper inhalation of low-tar, nicotine and carbon monoxide (TNCO) cigarette smoke [8]. Two consecutive processes determine the individual exposure of a smoker: the generation and emission of tobacco smoke during puffing, followed by inhalation of this emission by the smoker [3]. These two processes interplay and determine the intake of emitted smoke into the mouth cavity and lungs of the smoker. Interestingly, experienced smokers adjust their puffing topography, consciously or not, to modulate their nicotine, flavor and carbon monoxide (CO) exposure [9].

The puffing behavior relates to how a smoker smokes a cigarette and is characterized by parameters such as puff volume, puff duration, inter-puff interval and the number of puffs. Together, these puff parameters influence the airflow through the burning rod, which on its turn determines the combustion and pyrolysis rate of the tobacco and the physico-chemical processes inside the cigarette. For example, a high airflow induces higher temperatures and increases mass burn rate of the tobacco (mg/puff) [10,11]. Additionally, an increased puffing intensity (i.e., the product of puff volume and puff frequency) leads to more rapid burning of tobacco and less increased tar, nicotine and CO yields [12,13,14]. However, TNCO yields were shown to correlate linearly with total puff volume (i.e., the product of puff volume and puff number) [14]. Moreover, several studies using machine smoking have shown that varying puffing profiles lead to differences in volatile organic compounds (VOCs) [15], polycyclic aromatic compounds (PAHs) [16] and aldehydes [17] in cigarette emissions. For machine-smoked cigarettes, emissions of TNCO and other toxicants are at least two times higher when a Health Canada Intense (HCI) regime [18] was applied than when International Standardization Organization (ISO) 3308 regime [19] settings were used [20]. Among the different puff parameters, puff volume showed the greatest effect on carbonyl delivery [4]. Smoking behavior has been widely studied and was summarized for the last time in a report in 2000 [21]. Since 2000, Chen et al. published a study wherein a total puffing volume of 360 mL/cigarette was determined [22], i.e., the lowest value we could identify. This is almost four times lower than the 1289 mL/cigarette calculated with the data from Ross et al. and Farris et al. [23,24], i.e., the highest total puffing volume reported. The variation in puffing topography data published thus far may be explained by differences between cigarette brands as well as by inter-individual and intra-individual differences. Inter-individual differences in puffing topography are influenced by gender, ethnic background and type of cigarette. Indeed, it is known that women take smaller puffs of shorter duration but more puffs per cigarette than men [7], Korean-Americans, for instance, smoke with higher average puff flows and shorter inter-puff intervals than White-Americans [25], and that menthol and low-TNCO cigarettes are smoked with a higher puff volume and longer puff duration than regular cigarettes [26,27]. Furthermore, intra-individual differences are observed and smokers vary their smoking depending on the setting [28], cues [29], nicotine dependence, mood and emotional status [30]. In addition, there is variation throughout the course of the day as, for instance, the first cigarette in the morning is puffed less intensely [31], whereas smokers leaving work-stations to smoke outside buildings smoke their cigarettes nearly 19% more intensely than cigarettes smoked in social settings [32]. The research setting also affects smoking behavior, as smokers have more intense smoking behavior during lab visits than in a private setting [28]. To minimize the influence of brand or experimental setting, we have performed a human volunteer study in a real world setting with 25- to 34-year-old males to determine their puffing topography, when smoking only one brand of cigarettes ad libitum. Based on the observed puffing profiles of individuals, these personal smoking profiles were mimicked on a smoking machine and mainstream smoke TNCO levels were analyzed. With the results of the study, we aim to better understand the relevance of the machine smoking emission data (generated with ISO and Health Canada regimes) in relation to exposure of human smokers.

During analysis of the individual smoking topography data, we noticed unexpected outcomes in our study. Compared to the literature, a large number of puffs in combination with a high puffing volume were registered. Puffing topography was, like in many other studies, recorded with the CReSSmicro™ device. This device is described as a reliable and accurate instrument to record human smoking behavior [33,34,35]. However, we know from the literature that there is inconsistency in data treatment [36]. Moreover; malfunctions or anomalous data have been described; for example, high volumes (up to 5 L) [37] and a wide range of puffs (7–43 puffs) [38]. Furthermore, signal dropouts from unknown origin were found in the raw data of a study wherein the device was considered for measuring vaping behavior [39]. To investigate the origin of the unexpected data, we evaluated the CReSSmicro™ device in its performance to record human smoking behavior. 

## 2. Materials and Methods 

### 2.1. Literature Search

The last review listing smoking behavior originates from 1999 [9]. Therefore, an extensive literature search was performed for articles reporting puffing topography data that where published since 2000. The sources were the electronic databases PubMed, SCOPUS, EMBASE and WHO report series. Search terms included smoking topography, puffing topography, smoking behavior, puffing parameters, and smoker(s) or smoking in combination with human study. Reference lists were reviewed for additional references. Studies were included if puffing parameters were recorded by a desk or portable smoking topography device. Only puffing topography data of (mentally) healthy participants were included, and also when this was a control group. 

### 2.2. Recruitment of Participants

Five male participants were recruited by national social media. Only Caucasian/Europe-originated male smokers accustomed to using Marlboro red/regular cigarettes were included. Participants had to be used to smoking 13–25 cigarettes a day for at least 3 years. Participants were excluded if suffering from respiratory diseases or chronic illnesses, daily medication use, and experience of adverse effects due to smoking. The study was approved, according to the Declaration of Helsinki, by the accredited medical ethical committee (METC 153057) in Maastricht (The Netherlands) and registered online at ToetsingOnline (NL55676.068) (i.e., Dutch internet portal for the submission, review, registration and publication of medical research involving human subjects). Informed consent was signed before the experiment started and participation was rewarded with EUR 100.

### 2.3. Study Protocol

The participants stayed in an apartment for 36 h to create a homelike atmosphere, where they had breakfast, lunch, dinner, snacks and drinks ad libitum at their disposal. Participants arrived at 8 p.m. to sign informed consent and receive CReSSmicro™ usage instructions. Hereafter, participants could settle in the apartment and were allowed to smoke freely. The next morning, lighting the first cigarette of the day was noted as the start of the experimental day (t = 0). During the day, participants could smoke Marlboro cigarettes ad libitum using the CReSSmicro™. The experiment ended after smoking the last cigarette of the evening before going to sleep. The next morning, participants could leave. 

### 2.4. Cigarette Brand

The cigarette type used in the study was Marlboro red/regular king-size, since Marlboro is the most popular cigarette brand in the United States [40] and has the largest market share in the Netherlands, varying between 32% and 39% during the 2012–2017 period [41]. The researchers bought all cigarettes at a tobacconist in The Netherlands to pursue matching batch numbers. Cigarettes in the manufacturer’s unopened packaging were stored until their distribution to participants. Tar, nicotine and CO levels were 10, 0.8 and 10 mg/cigarette, respectively, as measured by the ISO method according to the package. 

### 2.5. CReSSmicro™ Analysis

Time of smoking was noted to give an overview of the natural smoking moments during the day. Puff parameters of all cigarettes smoked during the experimental day were monitored and recorded with the handheld, portable version of the clinical research support system (CReSSmicro™ v2.0.0; Plowshare Technologies, Baltimore, MD, USA) [34,35,42]. The device has a sterilized flow meter mouthpiece connected to a pressure transducer, which converts pressure into a digital signal that is sampled at 50 Hz. CReSSmicro™ computer software transforms the signal to a flow rate (mL/s) to compute puffing topography data. The three CReSSmicro™ devices used were calibrated according to procedures described in the manufacturer’s user manual. The calibration was verified at the end of every experimental day. The software of the CReSSmicro™, designed by Borgwaldt, uses the 50 Hz raw data to show a summary of puff profiles in the viewer of the program. A puff cleanup procedure (using the CReSS CleanUp program) was followed to make correct machine-generated artefacts in the data. In the case of inter-puff interval (IPI) <300 msec, the volume and duration were combined with the previous puff and remaining puffs with duration <100 msec or volume <5 mL were deleted as they are most likely noise from the machine. The puff parameters according to the software were presented as descriptive statistics (frequencies, means and standard deviations) for puff duration (sec), puff volume (mL), inter-puff interval (sec), flow per puff (mL/sec) and the number of puffs taken per cigarette per participant. The summarized data were used to calculate a personal puffing profile by linear regression of puff parameters versus the puff number, using a model with an intercept and slope for the overall data, as well as a first puff-specific parameter that allows for longer first puffs when lighting the cigarette. The raw data of the CReSSmicro™ device recordings were manually checked for anomalies such as flow rate dropouts or drops to zero (i.e., the record signal dropped to 0 mL/sec in the middle of a puff, and then went back up to the pre-dropout flow rate). 

### 2.6. Machine Smoking and Chemical Analysis

The calculated personal puffing profiles were mimicked on the smoking machine. Mainstream smoke of Marlboro cigarettes was generated with a 10-port linear smoking machine (SM410RH, Cerulean, United Kingdom) with Human Puff Profile Software and a rounded sinusoidal waveform in 100 Hz steps. Cigarettes were smoked according to the personal puff profiles to determine tar and nicotine (in fivefold) and CO (in twofold) as described in detail in ISO 4387:2000, ISO 8454:2007, and ISO 10315:2014 [21,43,44]. 

### 2.7. CReSSmicro™ Device Evaluation Experiment

An observatory experiment was performed to assess the impact of inserting a cigarette in the CReSSmicro™ device on flowrate recording. Based on the user manual, the insertion was guided with a beep to inform the smoker that the device is standing by and the smoking process will be recorded. We tested three methods of insertion; slow and loose insertion, normal insertion, and tight insertion whereby the cigarette is inserted as forcefully as possible without damaging it. The cigarettes were machine smoked with the CReSSmicro™ device positioned in the smoking machine, and in the adjacent port a cigarette was directly inserted into the cigarette holder of the smoking machine. The HCI regime [18] was set, which is a regularly used machine smoking regime (puff volume 55 mL, puff duration 2 sec, inter puff interval 30 sec). A second smoking protocol derived from averaging puffing parameters in the literature included a puff volume of 65 mL, puff duration 2 sec and inter-puff interval 18 sec. As an accuracy check, the recorded puff parameters by the CReSSmicro™ device were compared with the set smoking machine protocol. 

## 3. Results

The literature search resulted in 73 articles and reports specifying at least one or more puffing parameters. Appendix A in Appendix A lists per article or report, the participants’ gender, age, Fagerström index, cigarettes per day, cigarette type and recording device, if reported. The listed puffing parameters were puff volume, duration, flow, interval, count and total volume per cigarette and per day. If possible, missing puffing parameters were calculated. 

For the human study, the selected five male participants were 26 to 29 years old (mean 27.6 years). Based on self-assessment, they started smoking on average at the age of 16 years (10–20 years old) and had smoked on average 18 cigarettes per day (15–22 cig/day) since the age of 19 years (12–23 years old) (self-reported).

### 3.1. Smoking Timepoints

Smokers could smoke ad libitum during the experimental day, which resulted in an average of 17 cigarettes per day (10–21 cig/day) (Figure 1). Time between two cigarettes differed between 20 min and 3 h. For all participants, the second cigarette of the day was smoked within an hour after the first cigarette and the final cigarette of the day within an hour before going to bed.

### 3.2. Personal Puffing Profile

The smokers included in the study expressed no differences in enjoyment or satisfaction between smoking with or without the CReSSmicro™ device. The average puff parameters per participant, according to the puff profile software of the CReSSmicro™ device, are depicted in Table 1. The number of cigarettes smoked during the day and the puff parameters (puff count, puff flow, puff duration, and inter-puff interval) per cigarette varied considerably between smokers. The average puff count ranged between 11 and 26 puffs. This is at least two puffs more than we previously reported for machine smoking a Marlboro cigarette with smoking regimes of ISO and HCI [17]. Smoker 1 had the highest puff count (26 puffs) and a high puff volume (80 mL), which is much higher than machine smoking under ISO or HCI regimes. Smoker 3 also had a high number of puffs (17 puffs) and the highest puff volume (93 mL). Smokers 1 and 3 had a short smoldering period between two puffs (11 and 10 sec), attributing to more puffs in total and the highest total puffing volumes. These highest total puffing volumes (2127 and 1582 mL) are at least five times the machine smoking ISO (280 mL) and three times the HCI (495 mL) total cigarette volume.

The average puff volume (44 mL) of smoker 4 was in the range of the machine smoking regimes (35 and 55 mL). He had a short puff duration of 0.9 sec, which was half of the puff duration in the smoking regimes (2 sec). The longest puff duration was measured in smoker 5 (2.6 sec). As the puff volume (67 mL) and puff duration (2.6 sec) were higher than in HCI (55 mL and 2 sec), it still led to a comparable puff flow as HCI (27 mL/sec). The average puff flow of all smokers was at least twice the puff flow of the ISO regime, except for smoker 5. A substantial different puffing profile can still lead to a total puffing volume per day in the same range, as is seen for smokers 2, 3 and 5, with, respectively, a total puffing volume per day of 16.3, 15.8 and 14.8 L for all the cigarettes. However, as was shown by smokers 1 and 4, respectively, more than two times higher and lower total puffing volumes per day can also be observed.

### 3.3. Puff Profile over the Course of the Cigarette

The puff-by-puff pattern over the course of a cigarette smoked, averaged for all cigarettes used during the day, is shown in Figure 2. Participants smoked all cigarettes according to a characteristic puff profile that displays large differences between individuals. Smoker 1 typically inhaled almost the same volume each consecutive puff of a cigarette with limited variation between cigarettes (Figure 2, panel a). His puff flow increased slightly with consecutive puffs (Figure 2, panel b), whereas the puff duration shortened (Figure 2, panel c). Only the first puff of each cigarette of this smoker was longer than the other puffs with lower puff flow. Except for smoker 5, the other participants also increased the puff flow after the first puff. All participants, except for smoker 3, shortened the puff duration with consecutive puffs. Smoker 3 doubled his puff volume during the course of the cigarette smoking with a stable duration of subsequent puffs and increasing puff flow during smoking cigarettes. However, this smoker showed significant differences in the way he smoked the various cigarettes, which resulted in large standard deviation of puff volumes. During the smoking process, the inter-puff interval or smoldering period was almost stable for smokers 1, 2 and 5 and slightly increased for smokers 3 and 4. However, all smokers had a deviating longer first to second puff interval. In summary, individual participants smoked all cigarettes according to personal puffing topography, allowing the generation of a characteristic profile using linear regression of the puff parameters (Figure 2, dashed line). Four randomly chosen cigarettes per participant were sufficient to calculate this personal puffing profile per participant, as adding more or different cigarettes to this calculation did not significantly modify the created profiles.

### 3.4. Machine Smoking Puffing Profile

The puffing parameters of the personal puffing profiles were used as input smoking regime settings (human puffing regime) for the smoking machine. The smoking machine successfully finished the human puffing regime of smokers 2, 4 and 5. The total puffing volume with machine smoking was within 10% range of the measured total puffing volumes with the CReSSmicro™ device in the human study for smokers 2, 4 and 5. The human puffing regimes of smokers 1 and 3 consisted of an excessive number of puffs that was not possible to complete with the smoking machine as the tobacco was already completely burned before the total smoking protocol could be finished. This also explains why the total cigarette volume in machine smoking is lower than the total puffing volume measured with the CReSSmicro™ device for these smokers. The different human puffing regimes led to a range of TNCO yields in the smoke produced by the smoking machine (Table 2). Only the human puffing regime of smoker 4 showed TNCO yields within the range of HCI and ISO. When human puffing regimes of the other smokers were applied on the smoking machine TNCO yields were almost twice the ISO TNCO yields. For most smokers, TNCO yields were even higher than yields produced with the HCI regime. The tar and nicotine yields of the human puffing regimes of smokers 1 and 3 were approximately 70% more than the yield measured with HCI. CO yields were in the range between ISO and HCI, with an exception for the human puffing regime of smoker 1, which produced much higher yields.

### 3.5. Evaluation of CReSSmicro™ Device

Marlboro cigarettes were machine smoked with the CReSSmicro™ device according to two different regime settings. The HCI regime settings were used, and a more intense setting with a higher puff volume (65 mL) and a shorter inter-puff interval (18 sec). The latter settings were close to the mean values of the five participants as listed in Table 1. 

The two smoking machine regimes were combined with the three methods of insertion of the cigarette into the CReSSmicro™ device (Table 3). The CReSSmicro™ recorded the puff duration, which was similar to the smoking machine setting in all cases. However, the puff volume as well as the inter-puff interval was not similar to the machine settings in most cases. The puff volume registered by the CReSSmicro™ was higher in all cases, whereas the inter-puff interval was shorter. For example, a ~10% higher volume (59 mL) was measured with 55 mL puff volume after normal insertion with an inter-puff interval registered of 27 sec instead of 30 sec. 

The smoking machine generated 10–12 puffs to finish the cigarette directly inserted into the machine under smoking regimes (Table 3). For the cigarettes ‘tightly’ inserted into the CReSSmicro™ device and smoked on the smoking machine, a comparable number of puffs was observed for the two smoking regimes. For ‘normal’ or ‘loosely’ inserted cigarettes in the CReSSmicro™ device, however, up to 16 extra puffs were required to finish the cigarette on the smoking machine. 

Higher puff count, higher puff volumes and shortened inter-puff intervals resulted in a higher total smoke volume for all experimental conditions when the CReSSmicro™ device was combined with the smoking machine.

The flow-dropouts in the raw data were characterized by the recorded signal dropping to zero followed by the pre-dropout flow rate, which could occur at any time during a puff and was visualized in Figure 3. In the case of a normal insertion, an occasional flow-dropout (five per cigarette) was measured in a single puff whereas when the cigarette is loosely inserted, more flow-dropouts occurred (nine per cigarette). No flow-dropouts occurred after tight insertion into the device, which means there was no failure in the flowrate recording.

## 4. Discussion

### 4.1. Characterizing Human Smoking Behavior over the Course of the Day

Consumer exposure to smoke toxicants varies considerably due to differences in the emissions produced by different kinds of cigarette as well as differences in human smoking behavior. We measured puffing topography with the CReSSmicro™ device in smoking volunteers in a real-world situation. The puffing parameters of the participants were used as input for smoking machine experiments to determine TNCO emissions. 

Although some participants had a marginal variation in a specific puff parameter (e.g., participant 1 puff volume), overall, the puffing profiles of each cigarette smoked by the participants were not substantially different over the course of a day. This indicates that self-reported preferred cigarettes were not subject to change in puff parameters. This is in line with Hammond et al., who found a high degree of stability in puffing profile within subjects over time in three 1-week trials [45]. Interestingly, a study of Grainge et al. found a slightly lower intensity of smoking of the first cigarette of the day compared to other cigarettes during the day [31].

Few studies have focused on puff parameters during the course of smoking a cigarette, i.e., puff-by-puff [46,47,48]. We found a similar pattern as two other studies in adolescents, namely a decreased puff volume and duration, coupled with an increased puff flow and inter-puff interval [49,50]. An increased puff flow over the course of the cigarette can be explained by the draw resistance due to a reduction in the length of the cigarette after each puff. An explanation of why smokers change their puff parameters over the course of the cigarette might be that as the cigarette shortens, the delivery of nicotine and other smoke constituents, including tar and CO, increases per puff [47]. A decrease in the volume and duration of puffing may in turn bring about a consistent delivery of nicotine and CO throughout consecutive puffs of the cigarette. For all smokers, the first puff differs from the other puffs taken with a consequent different exposure content [51]. This deviating first puff is referred to as the lighting puff as smokers might adapt their puffing in such a way to only light the cigarette properly, and not necessarily to dose nicotine, and therefore take a puff much shorter than the consecutive puffs. We assume that a short lighting puff is followed by a larger second puff that is fully inhaled. Furthermore, some other smokers take a large first puff (up to 3.5 sec instead of less than 2 sec on average), whereby we assume that a large first puff is fully inhaled in addition to lighting the cigarette. In a puff-by-puff study, the initial lighting puff and puffs between the first three and last three puffs were deleted [49]. We suggest including all puffs because the exposure estimate depends on smoking the whole cigarette.

### 4.2. Puffing Topography in Literature and the Use of CReSSmicro™ Device

The individual puff parameter data of smokers 2, 4 and 5 of the present study are in line with the wide range of puff parameter outcomes reported by other studies (Appendix A in Appendix A), but not for smokers 1 and 3. Other studies vary in study design, setting, cigarette brand, changing cigarette characteristics over the years and study groups. Data reduction techniques and the impact of adjusting for puff count should be considered by interpreting different studies according to De Jesus et al., who systematically reviewed data produced with CReSSmicro™ [36].

In almost all studies, the puff duration is typically between 1 and 2 sec, although one study reported a shorter puff duration of 0.9 [33], similar to participant 4 of the study at hand (0.9 sec), and another study reported a longer puff duration of 3 sec [52]. Also, the lowest (31 mL [33]) and highest (85.1 mL [53]) puff volumes reported in the literature (Appendix A in Appendix A) comprise the puff volumes in our study. The puff flow reported ranges from 25.6 [54] to 62 mL/sec [53], which also comprises the puff flow in the present study (26–58 mL/sec). The average puff count reported in the literature (Appendix A in Appendix A) is between 7.6 puffs (a low-tar cigarette) [22] and 20.4 puffs [24]. 

We had a participant (smoker 1) with the extreme average puff count of 26. It was also this participant that had an extreme total puffing volume of 2127 mL, which is almost twice the highest total puffing volume reported in the literature (1451 mL/cig) [55]. For another participant (smoker 3), a very high puffing volume was also registered (1582 mL/cig), although the puff count was not high. Both smokers had a short inter-puff interval and a high puff volume registered by CReSSmicro™ device. When cigarettes were smoked with a large number of registered puffs in combination with a high puffing volume, a substantial number of flow-dropouts with the CReSSmicro™ device were observed. We randomly used three CReSSmicro™ devices following the instruction of the producer, and did not observe malfunctioning of individual devices.

We experimentally checked the outcomes of the CReSSmicro™ device data under controlled experimental conditions with a smoking machine and observed substantial differences between the CReSSmicro™ device data and the smoking machine settings. As was shown in the smoking machine experiment with the CReSSmicro™ device only flow-dropouts were registered for loose and normal insertions, and were very limited with tight insertion (Figure 3). Tight insertion was the insertion method that produced a similar number of puffs in the smoking machine for a cigarette with and without the CReSSmicro™ device. When ‘normal’ insertion was applied, the number of puffs per cigarette increased, and consequently the total puffing volume. When ‘loose’ insertion was applied, the puff number more than doubled and puffing volume became extremely high, particularly when a high puff volume and short inter-puff interval was applied. This suggests that in the case of a loose or even normal insertion of cigarettes in the CReSSmicro™ device, in the human smoking experiment the smokers may depend on the insertion method, leading to the recording of puffing data which are a pitfall of the methodology. In our study, participants inserted the cigarettes themselves and did not indicate loose, normal or tight. However, they only followed the user manual of the device, and the researchers observed no deviating behavior. The summarized data in the software of the device gave no alert of incorrect or incomplete puff parameters, other than the flow-dropouts. Flow dropouts may, however, indicate that the device is not functioning properly. The recording of dropouts is also reported in the literature, but to the best of our knowledge no information is available whether or not data about ‘high’ puffing volumes, or ‘high’ puff numbers, have been deleted in previous studies. 

During the machine smoking experiments, smoke was observed at the cigarette insertion opening in the device (Appendix A in Appendix A). During the smoking machine experiment, smoke at the mouth end of the cigarette was also observed, especially during the puff, and shortly after the puff. Human smoking with a loose insertion may lead to sidestream air entering the cigarette at the mouth end of the cigarette or air may be drawn directly into the device. In addition, as the filter (paper) is permeable at the mouth end of the cigarette, it may be possible that air is entering the device without passing or going through the cigarette, thereby diluting the smoke. On the other hand, it is also possible that smoke is leaking before entering the device. This was not mentioned by the participants nor observed by the researchers during our study. In the case of false air entering the device, we hypothesize that less tobacco is combusted with every puff, which makes it possible to take many puffs in total to finish the cigarette. If this artefact exists for a particular participant smoking a cigarette, he will try to inhale a large puff volume to satisfy his nicotine intake, probably in combination with many puffs. Due to the dilution of the smoke entering the smoker’s mouth and the low amount of tobacco burnt, the nicotine intake per puff will be low. The desired effect of nicotine is not attained because of this dilution, motivating the smoker to take larger puff volumes and more puffs than he would take without the device. 

### 4.3. TNCO Yields Produced by Machine Smoking with Personal Human Puffing Regximes, ISO Regime and HCI Regime

There are three studies that had elements as in the present study. First, Hammond et al. recorded the puffing profile of 51 participants while smoking their usual brand; subsequently, the average puff volume and average puff frequency were used as parameters for machine smoking instead of the puff-by-puff profile as used in the study at hand [20]. Second, Djordjevic et al. recorded puffing profiles of 133 smokers and 72 randomly chosen puffing profiles were used for smoking machine setting for the medium-yield cigarettes [56]. Only averaged TNCO yields are reported and these are not linked to puffing parameters. In general, the literature data are averaged among study participants and almost no data from individuals are available. This hinders the comparison of study outcomes and identifying ‘extreme’ intense smoking. The third study, Dickens et al., is the only study found that reported individual puffing parameters (*n* = 7), but they were not replicated with a smoking machine [57]. These participants also had intense puffing profiles with puff volumes in the range of 55 to 119 mL per cigarette, with puff durations of 1.5 to 3.3 sec [57].

In our study, smoking machine experiments were used in a laboratory setting to determine cigarette smoke toxicant yields that are produced by puffing according to specified settings that were derived from the human smoking experiment. The linear regression (described in 2.5) showed that data of four cigarettes for a particular participant are sufficient to calculate the smokers’ personal human puffing regime. When the human puffing profiles of smoker 1 and smoker 3 were used for machine smoking settings, the cigarette was finished before the puff profile ended. In spite of this, these two human puffing regimes generated the highest TNCO yields. Since the recording of profiles with CReSSmicro™ device may have overestimated the actual inhalation, the data of these two smokers are not further taken into account. 

The least intense puffing profile (based on data of smoker 4, i.e., lowest puff volume and total cigarette volume, shortest puff duration and longest inter-puff interval) generated the smallest TNCO content. Hammond et al. reported less intense puff parameters (puff volume 53.3 mL, inter-puff interval 33.2 sec, puff duration 1.4 sec) than in the present study, and this might explain why they found lower TNCO values (tar 26.7 mg, nicotine 2 mg, CO 24.6 mg) for comparable ‘regular yield’ cigarettes (9–15 mg ISO tar) [20]. In comparison with our puffing profiles of smokers 2, 4 and 5, the participants in the study of Djordjevic et al. had lower (total) puff volumes (523–615 mL versus 495–778 mL in our study) and a longer inter-puff interval (19–21 sec versus 16–21 sec in our study) (similar puff duration (1.5 sec) and puff count (12 or 13 puffs)) [56]. Our data, as well as data collected from the literature, indicated that the inter-puff intervals applied for ISO and for HCI (respectively 60 and 30 sec) are too long to represent human smoking. This leads to long smoldering and lower puff numbers when ISO and HCI regimes are applied.

In our study, tar yield was on average 29 mg/cig (21–38 mg/cig), nicotine 2.39 mg/cig (1.5–2.4 mg/cig) and CO 22.5 mg/cig (16–20 mg/cig). When TNCO yields generated according to human puffing regimes are compared to smoking machine generated emissions according to ISO and HCI settings (Table 2), it is clear that both ISO and HCI produce fewer puffs per cigarette. The HCI generates yields for TNCO which are in the same range yields produced with the human smoking regimes of smokers 2, 4 and 5. This is consistent with previous studies [20]. On average, the puff parameters of Hammond et al. [20] are more comparable with the HCI regime than those of the present study. They found that their human mimic regime on the smoking machine produced slightly less nicotine (2 vs. 2.4 mg/cig), tar (24.7 vs. 30.6 mg/cig) and CO (24.6 vs. 28.1 mg/cig) than the HCI regime, while they observed a larger total cigarette volume (53.3 mL × 11.5 puffs = 612.95 mL) and higher puff flow (38.6 mL/sec) [20]. It was also clear that ISO regime underestimates human smoking substantially [20]. The TNCO yields determined when using the human puffing profile regime were at least twice as high as the yields we determined with ISO [17] regimes on the same smoking machine. The total puff volume of 280 mL is very low compared to data reported in the literature, as well as the data generated in the study at hand. The lowest total puffing volume reported in the literature is 380 mL, which is close to that of the ISO regime for smoking machines [22]. A higher volume per puff in combination with a higher puffing frequency better mimics human smoking intensity. 

The proxy of number of smoked cigarettes per day is often used in risk assessment modelling when estimating the dose of exposure to toxicants in smokers [7,58]. However, as is shown in this study, this proxy has limitations for actual risk assessment of cigarettes since it does not correspond with total puffing volume per day. Others state that smoke volume correlates well with biochemically assessed human smoke exposure [31,45] and that total puff volume and duration measured with the CReSSmicro™ device are likely to give the best approximation for toxicant exposure estimates [36]. Moreover, the inhalation of the cigarette smoke ultimately leads to systemic exposure. The puff parameters give an indication of total smoke volume and depth of inhalation [26], which provide information on which part of the lungs is reached by the smoke toxicants. To estimate the final systemic exposure to cigarette smoke toxicants, we need to relate smoke content, puffing parameters and respiratory parameters, especially inhalation volume and duration. Knowledge about cigarette smoke inhalation parameters is limited and the realistic human puffing profiles and TNCO emissions from the present study can improve the predictions for (site-specific tissue dose) computer models [59]. Studies often focus on the influence of a single determinant, i.e., age [49,50], gender [7], racial background [25,60,61], or cigarette brand [27,62] on smoking behavior in healthy smokers [33,63]. The present study took place in a controlled, but not entirely artificial, real-world environment that accommodates the smoking experience [48,61,64]. In the small but homogenous group, of smokers’ inter-individual differences in the number of cigarettes smoked, the time points of smoking, and the mean and total puff parameters were registered. This suggests that estimating overall exposure based on smokers’ characteristics is not sufficient and that variations of smoking topography contribute to the smoker’s risk. The limitation of the study at hand is that the current sample of smokers and cigarette brand is not representative for all smokers or cigarette brand. Moreover, other determinants might cause even more variations in cigarette emissions and exposures of the smokers.

## 5. Conclusions

Smokers smoke their cigarettes with consistent, individually characteristic puffing topography. These characteristic human puffing profiles show differences between smokers. The variety in puffing profiles within the homogenous study group of the present study was also seen in the compiled literature data of puffing topography studies since 2000 (Appendix A in Appendix A). When the different characteristic human puffing regimes are mimicked on the smoking machine, differences in TNCO yields are generated. Comparison with machine smoking data shows that smokers are likely to be exposed to at least twice the TNCO yields as measured with ISO regimes on a smoking machine. This observation indicates that the ISO 3308 regime largely underestimates human exposure due to low puffing intensity. Smokers’ exposure to TNCO can probably be better estimated with machine smoking set on the HCI regime. However, the present study shows that the smoking machine with the HCI regime may also underestimate actual smokers’ exposure, because inter-puff intervals are shorter for human smokers than in the HCI regime. 

## Figures and Tables

**Figure 1 ijerph-17-03225-f001:**
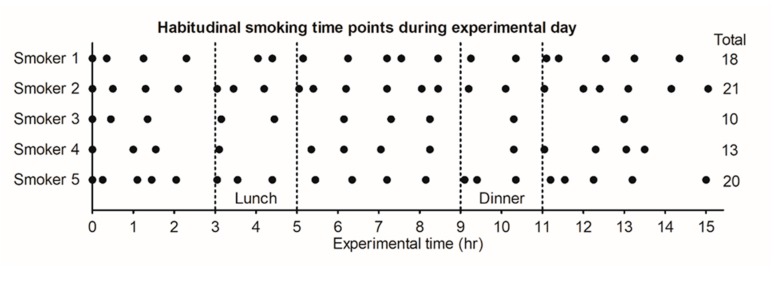
Smoking time points and total of cigarettes smoked by participants during the experimental day, describing their daily smoking behavior. Lighting the first cigarette of the day was defined as t = 0.

**Figure 2 ijerph-17-03225-f002:**
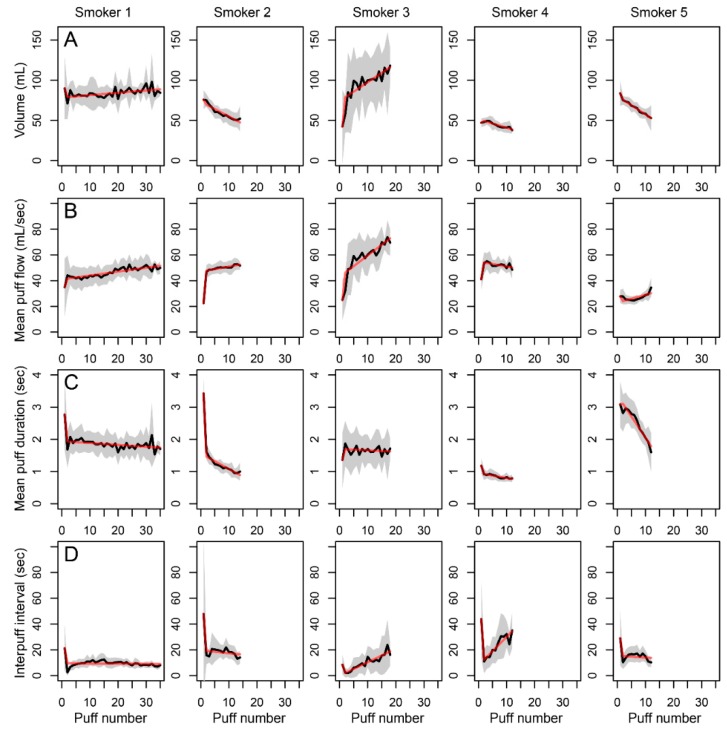
Individual puffing profile consisting of volume (mL), average flow (mL/sec), duration (sec) and IPI (sec) of subsequent puffs (puff number) per smoker. Shown is the mean of all cigarettes smoked during the study day (black line) with SD (grey area), whereby the dashed line is the model fit.

**Figure 3 ijerph-17-03225-f003:**
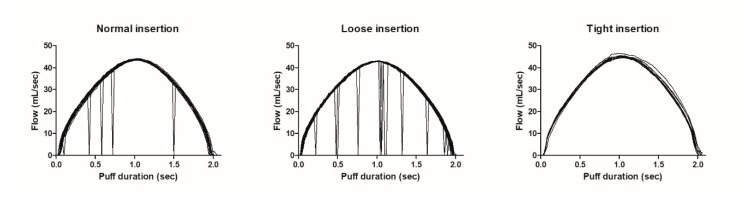
Example of flow-dropouts in raw data collected with the CReSSmicro™ device, when recording the smoking (HCI regime) of an entire cigarette after different ways of inserting the cigarette. Every line represents one puff.

**Table 1 ijerph-17-03225-t001:** The puff parameters per participant and smoking regime.

Participant or Smoking Regime	Cigarette Count	Puffs per Cigarette	Puff Volume (mL)	Puff Flow (mL/sec)	Puff Duration (sec)	Inter-Puff Interval (sec)	Total Puffing Volume per Cigarette (mL)	Total Cigarette Volume per Day (L) ^&^
	^#^	Mean (SD)	Mean (SD)	Mean (SD)	Mean (SD)	Mean (SD)	Mean (SD)	
Smoker 1	18	26 (8.9)	80 (11.4)	43 (7.7)	1.9 (0.2)	11 (2.4)	2127 (904)	38.3
Smoker 2	21	13 (1.3)	60 (5.7)	48 (3.0)	1.4 (0.1)	21 (5.8)	778 (123)	16.3
Smoker 3	10	17 (2.3)	93 (13.9)	57 (7.3)	1.6 (0.1)	10 (4.6)	1582 (296)	15.8
Smoker 4	13	11 (2.3)	44 (3.8)	51 (4.0)	0.9 (0.1)	26 (8.1)	495 (125)	6.4
Smoker 5	20	11 (1.0)	67 (4.1)	27 (2.3)	2.6 (0.2)	16 (3.1)	740 (79)	14.8
ISO *		8	35	17.5	2	60	*280* ^$^	
HCI *		9	55	27.5	2	30	*495* ^$^	

* Data originate from Pauwels et al. [17]. (Marlboro red according to ISO [19] and HCI [18]) ^#^ Cigarettes smoked during experiment. ^$^ Puff count multiplied with puff volume. ^&^ Total cigarette volume multiplied with cigarette count. The italics represent calculated numbers.

**Table 2 ijerph-17-03225-t002:** Puff count per human puffing regime during machine smoking with associated total puffing volume, yield of TNCO (mean (SD)).

Participant or Smoking Regime	Puffs Smoking Machine (SD)	Total Puffing Volume mL/cig (SD)	Tar mg/cig (SD)	Nicotine mg/cig (SD)	CO mg/cig (SD)
Smoker 1	12 (1.4)	1005 (116)	58 (9.6)	3.4 (0.4)	49 (0.8)
Smoker 2	12 (0.9)	710 (45)	32 (1.1)	2.1 (0.1)	18 (0.2)
Smoker 3	14 (1.0)	1200 (108)	59 (8.1)	3.3 (0.3)	27 (3.3)
Smoker 4	12 (0)	530 (0)	21 (1.4)	1.5 (0.1)	16 (^&^)
Smoker 5	11 (0.7)	754 (38)	38 (1.5)	2.4 (0.1)	20 (0.1)
ISO *	8	280 ^$^	10.37	0.8	8.51
HCI *	9	495 ^$^	34.04	1.97	26.27

* Data originate from Pauwels et al. [17]. (Marlboro red according to ISO and HCI). ^&^ failed duplicate. ^$^ calculated by puff count multiplied with puff volume.

**Table 3 ijerph-17-03225-t003:** Puff parameters (mean) recorded by the CReSSmicro™ device while smoking via two regimes on the smoking machine. Cigarette insertion into the device in a ‘normal’, ‘loose’ and ‘tight’ way.

Puff Parameter	Smoking Machine or CReSSmicro™ Device	Way of Cigarette Insertion in CReSSmicro™ Device					
		Normal		Loose		Tight	
		Regime 1	Regime 2	Regime 1	Regime 2	Regime 1	Regime 2
Puff count	Smoking machine	11	11	10	10	11	12
	CReSSmicro™ device	18	20	18	26	11	13
Puff volume (mL)	Smoking machine	55	65	55	65	55	65
	CReSSmicro™ device	59	73	60	74	65	76
Puff duration (sec)	Smoking machine	2	2	2	2	2	2
	CReSSmicro™ device	2	2	2	2	2	2
Inter-puff interval (sec)	Smoking machine	30	18	30	18	30	18
	CReSSmicro™ device	27	16	29	16	27	16
Total puffing volume (mL)	Smoking machine	990	1300	1008	1690	587	845
	CReSSmicro™ device	1053	1453	1106	1932	692	983
Flow-dropouts	CReSSmicro™ device	5	6	8	6	0	1

Regime 1: HCI settings (puff volume 55 mL, twice per minute for 2 sec). Regime 2: puff volume 65 mL, 18 sec inter-puff interval and 2 sec duration.

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
