# Peer review of "Characteristic Human Individual Puffing Profiles Can Generate More TNCO than ISO and Health Canada Regimes on Smoking Machine When the Same Brand Is Smoked"

_ijerph, 2020, doi:10.3390/ijerph17093225_

Round 1
Reviewer 1 Report
The manuscript titled “Consistency of individual puffing topography and differences between smokers cause substantial differences of TNCO exposure when the same brand is smoked” (ijerph-746963) reports puffing profiles of individual users in an ad-lib session of a whole day. While I think this study is important and makes it obvious that the ISO protocol may be insufficient, there seems to be a disconnect between the finding of the authors that the used cress device may be unreliable at certain times and the reporting of the findings themselves. I will provide more details below, but overall I would recommend to reconsider the manuscript if the authors are willing to make edits to it.
- Currently, the manuscript follows the following flow: first, the very different puffing profiles observed are reported, then a section on the cress device follows, including its shortcomings. Yet the authors do not seem to go back to evaluate the results obtained in the first section. The elephant in the room does not seem to be addressed: if the authors are not convinced that the device produces reliable data, are they certain of the results reported in the earlier part? For example, for smoker 1, the smoking machine could not replicate the user behavior because the cigarette burned out before the full number of puffs could be taken. Are the authors sure that the cress device worked correctly for this user? Also, the reported mismatch data between smoking machine and cress device (Table 3) is not taken into account for the participant results, correct? Was there an attempt by the authors to correct some of the data taken with the cress device by the results shown in Table 3?
- The authors included a large and systemic literature review (Supplementary file A), yet its existence is barely mentioned in the manuscript; possibly only in one sentence in the Introduction section. I would strongly recommend including in the Methods section how the data was collected (search terms, inclusion criteria, etc) and to also include a section in the discussion about this large set of valuable data.
- The authors should consider including in the Introduction or Discussion section a paragraph on the why the CHI protocol was developed in the first place? My assumption is that Health Canada similarly found discrepancies between the ISO protocol and “real” data. This is directly relevant to this study and should be included.
- Authors should consider including a Limitations section that could address the mismatch between cress and smoking machine data as well as the very small number of experiments (n=5).
- 56: authors use the term “puffing intensity” here and throughout the manuscript, yet it is never defined. The puffing parameters are defined, but never the “puffing intensity”. Please include this to make it crystal-clear what is meant.
- 83-99: Authors should consider if this section should be part of the Introduction?
- 122: “a large market share” seems vague
- 5-7: Authors should review the use of present vs. past tense here. Most sentences should be in past tense.
- Table 1: The puff flow for the ISO protocol seems too high (21.5 ml/sec)
- 230-232: How did the authors determine this bold statement? How was this obtained, is there some data to prove this?
- L300-302: “puff parameters were not sign. Different over the day”. How does this match with the large SD for smoker 1?
- 318: How did the authors determine which puffs were inhaled, and which were not? It does not seem that the cress device can measure this, so I can’t follow how the authors could have determined this.
- 326: “might be a reason for this”. Please clarify “this”
- 328-333: Since the authors only have data on 5 participants, how relevant is the reported data (and should that not be in the Results section?)
- L362-362: What is meant by “the human study”? Your study?
- 379: Authors mention a mathematical model here. Where is the model, how does it work, what are its limitations, etc?
Reviewer 2 Report
There are differences in human behavior in relation to puffing topography parameters which impacts their exposure to toxicants such as tar, nicotine and carbon monoxide (TNCO). This paper’s stated aim is to identify these variations for risk assessment.
Major Comments:
- Unclear Objective:
- The objective of the study is unclear. The objective was stated to be identification of variations in topography, but the discussion and conclusions are based on differences in regimes.
- Research Design:
- The sample size is extremely small.
- The sample is not representative of smoking population.
- Voluntary participation in the study further complicates selection issues due to small, non-representative sample.
- Policy recommendations:
- The authors do not explicitly discuss how the results can be utilized in policy making.
Round 2
Reviewer 1 Report
The authors have adequately addressed the concerns raised by both reviewers and the manuscript should be publishable in the current format.